# Hierarchical Prototype-based Explanations

## Abstract

To interpret deep neural networks, one main approach is to dissect the visual input and find the prototypical parts responsible for the classification. However, existing methods often ignore the hierarchical relationship between these prototypes, and thus can not explain semantic concepts at both higher level (e.g., water sports) and lower level (e.g., swimming). In this paper inspired by human cognition system, we leverage hierarchical information to deal with uncertainty stemming from a lack of information: When we observe water and human activity, but no definitive action, it can be recognized as the water sports parent class. Only after observing a person swimming can we definitively refine it to the swimming action. To this end, we propose HIerarchical Prototype Explainer (HIPE) to build hierarchical relations between prototypes and classes. HIPE enables a reasoning process by dissecting the input video frames on multiple levels of the class hierarchy. The faithfulness of our method is verified by reducing accuracy-explainability trade off on ActivityNet and UCF-101 while providing multi-level explanations.

## 1 Introduction

When describing the world around us we may do so at different levels of granularity, depending on the information available or the level of detail we intend to convey. For instance, a video might open with a shot of a cheering crowd, allowing us to recognize it as a *a sports event*, as the camera then pans to the river we can deduce that it is *a water sports event*. However, only when the raft comes into the frame can we determine that it concerns *rafting*. In resolving this uncertainty we navigate between levels of a hierarchy based on the information available. Nonetheless, in our description of this video, we may still only refer to it as a sports or water sports event. Yet, our reasoning and description processes build on the hierarchical relation between classes, allowing for navigation between generic and specific. In this work, we implement this process for video action recognition by learning hierarchical concepts that we leverage for explanations at multiple levels of granularity which also leads to improved classification performance, see Figure 1.

Despite the remarkable performance of neural networks for video understanding tasks (Hara et al., 2018; Karpathy et al., 2014; Peng & Schmid, 2016; Singh et al., 2017; Simonyan & Zisserman, 2014; Wang et al., 2016; Carreira & Zisserman, 2017; Bertinetto et al., 2016) it is still hard to explain the decisions of these networks. This has led to a growing stream of research that focuses on making models interpretable besides performing accurately (Hendricks et al., 2018; Kim et al., 2018b; Gulshad & Smeulders, 2020; Trinh et al., 2021; Li et al., 2021). A promising line of posthoc explainability methods are the concept bottleneck models Koh et al. (2020); Losch et al. (2019); Kim et al. (2018a), which focus on explaining decisions of neural networks by predicting human understandable concepts before performing final prediction. However, these methods require dense concept annotations or access from other resources like pre-trained language models Yuksekgonul et al. (2022); Oikarinen et al. (2023). Our method for providing builtin explanations does not require dense annotations to provide explanations at multiple levels of granularity.

In a similar spirit, prototype-based models Li et al. (2018); Chen et al. (2019); Donnelly et al. (2022) focus on learning prototypes and making predictions through case-based reasoning over these prototypes. This enables *this look like that* explanations. However, existing case-based reasoning works in the visual domain are limited to 2D models. Moreover, they provide a single level of explanation which in case of uncertainty

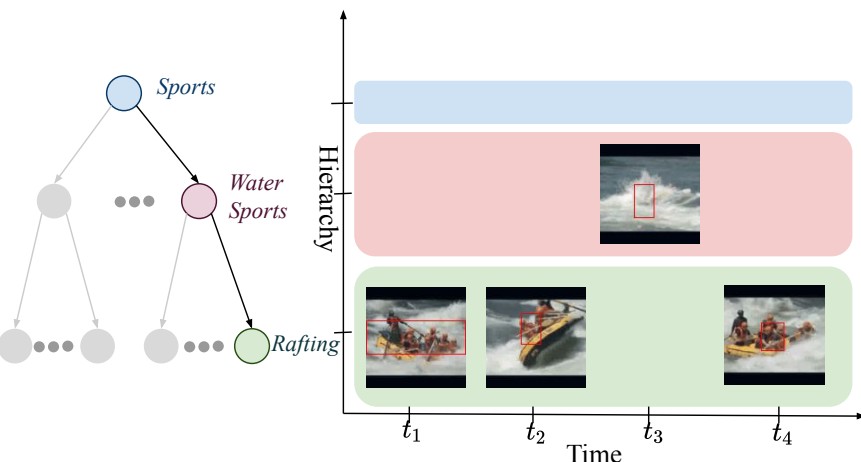

Figure 1: HIPE generates explanations by learning prototypes at different hierarchy levels, allowing it to produce meaningful predictions even when there is insufficient information. For example, at $t_3$ there is insufficient visual information to recognise it as rafting, however, it is still recognise as a water sport.

can be as bad as arbitrary, as each explanation is considered equally apart. In this work, we focus on capturing the hierarchical relations between actions to provide multi-level explanations for videos.

A challenge for explainable models, such as concept bottleneck models and prototype-based models, is that it introduces an accuracy-explainability trade off, where explainability comes at the cost of accuracy. With this paper, we aim to introduce a model with built-in explainability across multiple levels of granularity which we believe will be less affected by this trade off. This belief is inspired by recent works on learning hyperbolic embedding spaces, as opposed to euclidean, in natural language processing Tifrea et al. (2018b); Dhingra et al. (2018) and computer vision tasks Atigh et al. (2022); Ghadimi Atigh et al. (2021); Long et al. (2020). These works have demonstrated that it is beneficial for performance to have the embedding space be guided by hierarchical knowledge, which we believe will in turn also benefit explainability. As it will better match the hierarchical cognition process of humans, that is we are likely to organize concepts from specific to general Warrington (1975); Minsky (1982); McClelland & Rogers (2003), and the representation of categories in the hyperbolic space.

The main contributions of our paper are: 1) We propose HIerarchical Prototype Explainer (HIPE), a reasoning model for interpreting video action recognition. 2) We demonstrate that HIPE can provide meaningful explanations even in the case of uncertainty or lack of information by providing multi-level explanations i.e., at class, parent, or grandparent level. 3) We perform a benchmark and show that HIPE counters accuracy-explainability trade off.

## 2 Related Work

### 2.1 Interpretations for Videos

Interpretations for neural networks can be broadly classified into two categories: 1) fitting explanations to the decisions of the network after it has been trained i.e. *posthoc* Hendricks et al. (2018); Kim et al. (2018b); Gulshad & Smeulders (2020); Ribeiro et al. (2016); Lundberg & Lee (2017), 2) building explanation mechanism inherent in the network i.e. *built-in* explanations. Trinh et al. (2021); Li et al. (2021); Lin et al. (2021); Uchiyama et al. (2022). In this work, we focus on learning semantic representations which are used for classification during training rather than explaining a black box network posthoc.

A great deal of previous work has focused on video action recognition, detection, segmentation and more Hara et al. (2018); Karpathy et al. (2014); Peng & Schmid (2016); Singh et al. (2017); Simonyan & Zisserman (2014); Wang et al. (2016); Carreira & Zisserman (2017); Bertinetto et al. (2016), however, most of these

works focus on designing black box models for specific tasks. They do not explain why a certain decision is made by the model. Moreover, most of the research in the domain of visual explanations focuses on images. Only a few works focus on the interpretation of these networks for videos Karpathy et al. (2015); Bargal et al. (2018); Stergiou et al. (2019b); Li et al. (2021); Stergiou et al. (2019a), and it is not possible to directly apply image-based explanation methods to videos due to an extra time dimension in videos.

Karpathy et al. (2015) and Bargal et al. (2018) focus on visualizing spatio-temporal attention in RNNs, CNNs are used only to extract features. Inspired by class activation maps (CAM) Zhou et al. (2016) for images Stergiou et al. (2019b) extended it for videos by finding both regions and frames responsible for classification. Li et al. (2021) utilized perturbations to extract the most informative parts of the inputs responsible for the outputs. Both Stergiou et al. (2019b); Li et al. (2021) are posthoc methods, which means they do not use explanations during prediction therefore they might not be faithful to what the network computes Donnelly et al. (2022). Stergiou et al. (2019a) introduced class feature pyramids, a method that traverses through the whole network and searches for the kernels at different depths of the network responsible for classification, therefore this method is computationally expensive. In contrast, we enable built-in multi-level explanations that do not add any computational complexity. In this paper, we enable multi-level explanations for videos by learning hierarchical prototypes for each class and tracing them back to input videos.

## 2.2 Case-based Reasoning Models

There are two main categories of case-based reasoning models: *concept bottleneck models* which introduce a bottleneck layer that learns human understandable concepts, and *prototype-based models* that learn prototypes that are closer to the samples in the training set. Concept bottleneck models provide posthoc explanations by replacing the final layer of the neural network with a layer that predicts human understandable concepts e.g. for a cardinal bird class the concepts will be red wings, red beak and black eye Koh et al. (2020); Losch et al. (2019); Kim et al. (2018a); Wang et al. (2022). These predicted concepts are then used to perform classification. However, concept bottleneck models require dense concept annotations for the model to learn them. Moreover, as they use predicted concepts to perform classification therefore, they suffer from explainability-accuracy trade off. Yuksekgonul et al. (2022); Oikarinen et al. (2023) focus on addressing these limitations by either incorporating concepts by transferring them from natural language descriptions or generating them from a GPT model. In contrast, our work focuses on providing built-in explanations by learning representative samples for each class and its (grand)parent class, while countering explainability-accuracy trade off. Our method do not require heavy annotations but utilize either the hierarchy available with the datasets or it can be easily defined based on the relations between classes.

More relevant to our method, Wang et al. (2022) build relations between neurons and hierarchical concepts. However, our method differs from Wang et al. (2022) in three significant ways 1) their method is posthoc (a separate concept classifier is trained) while ours is builtin, 2) to learn hierarchical concepts they utilize wordnet hierarchy of Imagenet while we do not, 3) considers the contribution of neurons (non comprehensible for humans), whereas we visualize learned features by the prototype layer.

The idea behind prototype-based models, to provide built-in explanations with prototypes was first explored in Li et al. (2018), where the authors introduced a prototype layer in the network with an encoder-decoder architecture. The prototype layer stores weights which are close to encoded training samples, and a decoder is used to visualize them. However, their model fails to generate realistic visualizations for natural images. Thus Chen et al. (2019) proposed to learn prototypes for each class and visualized them by tracing them back to the input images without a decoder. We get inspiration from Chen et al. (2019) to provide built-in explanations, but where their work is limited to 2D images and provides only one-level explanations we extend it to multi-level explanations for videos.

Rymarczyk et al. (2020) focuses on reducing the number of prototypes for each class by finding shared prototypes among classes. Wang et al. (2021) introduced a different similarity metric for computing similarities between prototypes and image patches. Deformable ProtoPNet Donnelly et al. (2022) learns spatially flexible prototypes to capture pose and context variations in the input. All previous prototype-based explanation methods provide explanations without considering the hierarchical relations between classes on well-defined image CUB birds Wah et al. (2011) and Stanford cars Krause et al. (2013) datasets. In contrast inspired by

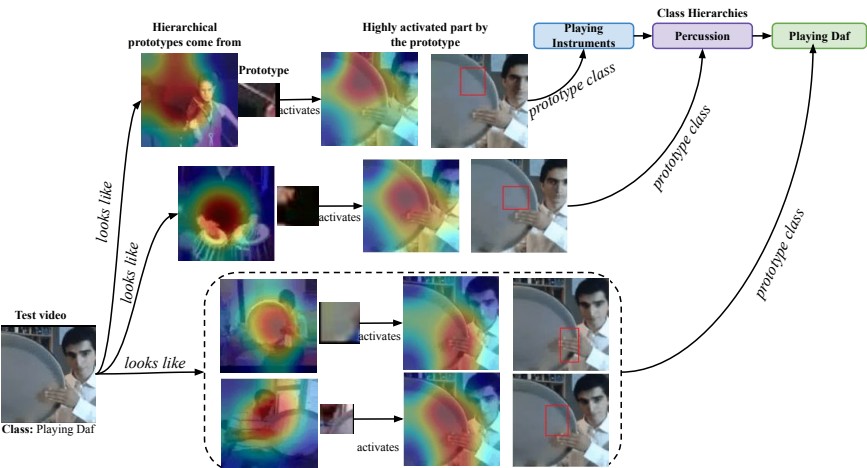

Figure 2: **Significance of Hierarchical Explanations.** Hierarchical explanations generated for an input video of playing daf. Even if someone does not know what is *"playing daf"*, its parent explanation shows that it involves *"percussion"* and grandparent shows that it is a musical *"instrument"*.

the human way of explanations we consider hierarchical relations between classes while learning prototypes for each class for video datasets.

Closely related to our work Trinh et al. (2021) introduced a dynamic prototype network (DPNet) for finding temporal artifacts and unnatural movements in deep fake videos. However, the goal of DPNet is different from ours with only two target classes fake/real making the task easier. PrototypeTrees Nauta et al. (2021) use a decision-tree with a pre-defined structure, where individual prototypes are learned at each decision. The prototypes are optimised to increase purity along the path through the tree. However, for PrototypeTrees the position in, and order of, the tree does not describe a hierarchy, that is closer to the root does not imply a more general semantic level. Moreover, as the number of prototypes depend upon the size of the tree, learning a ProtoTree becomes computationally expensive. Our proposed multi-level explanations follow a very clear semantic distribution , where (grand)parent prototypes are more generic and do not add any extra computational complexity.

Akin to our work, Hase et al. (2019) organizes the prototypes hierarchically and classifies objects at every level in a predefined taxonomy. They do so for only 15 classes from ImageNet Deng et al. (2009) by learning conditional distributions at each node. The main difference between HPnet and our HIPE lies in the embedding space where hierarchies are learned, HPnet learns hierarchical prototypes in the euclidean space while we learn them in the continuous hyperbolic space. Using hyperbolic space allows us to embed hierarchies with minimal loss of information Ganea et al. (2018); Sala et al. (2018), thereby scaling to much larger hierarchies (15 in HPnet versus 200 in our case).

### 2.3 Hyperbolic Embeddings

Hyperbolic embeddings have recently received increased attention as they enable continuous representations of hierarchical knowledge Nickel & Kiela (2017b); Chami et al. (2019). This continuous nature makes it such that information of (grand)parent classes is implicitly included, allowing hyperbolic training to remain single-label. Their effectiveness has also been shown for textual Tifrea et al. (2018a); Ganea et al. (2018); Zhu et al. (2020) and visual data Khrulkov et al. (2020); Atigh et al. (2022); Ghadimi Atigh et al. (2021); Long et al. (2020). Hyperbolic embeddings have also been used for zero-shot learning Liu et al. (2020); Fang et al. (2021) and for video action recognition Long et al. (2020); Surís et al. (2021). The hierarchical relationship between videos and the hierarchical way of explaining decisions for humans calls for the need of using hyperbolic spaces. Here, we utilize hyperbolic embeddings for learning hierarchical prototypes to provide human-like explanations for video action recognition.

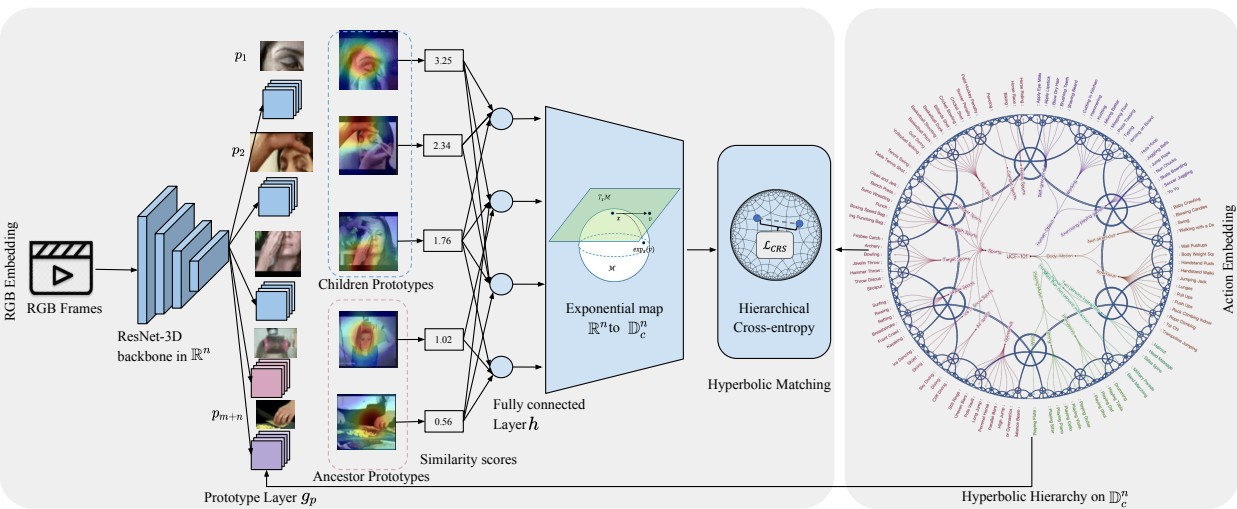

Figure 3: **Overview of the Hierarchical Prototype Explainer.** The Resnet-3D backbone extracts video features and the prototype layer learns prototypes for children and parents, these prototypes are then converted to a single similarity score through max pooling. Finally, scores are converted from $\mathbb{R}^n$ to $\mathbb{D}^n$ through a fully connected layer followed by an exponential map, to the shared hyperbolic space for hierarchical learning. Actions are mapped onto the shared hyperbolic space by learning a discriminative embedding on $\mathbb{D}^n$.

## 3 Hierarchical Explanations

Single level explanations are not always sufficient to explain the decisions of deep neural networks e.g., in Figure 2, if we only provide reasoning behind the prediction *"playing daf"* at a single level and someone who is not familiar with a *daf* may not understand this explanation. Instead, we provide explanations at more abstract levels i.e., parent and grandparent level. By looking at the explanations for parent class one can infer that it involves *"percussion"* and from the grandparent that it is some sort of musical *"instrument"*. We propose to incorporate this powerful mechanism for providing explanations based on hierarchical prior knowledge into networks for video action recognition.

### 3.1 Hierarchical Action Embeddings

Incorporating the prior hierarchical knowledge about actions into the network requires that we represent them as embeddings. In this section we detail how to learn those action embeddings in hyperbolic space, in the next section, we explain how to learn hierarchical prototypes that are optimized by aligning them to the action embeddings in hyperbolic space. Given the set of action classes $\mathcal{A} = \{1, 2, ..., |\mathcal{A}|\}$, in hierarchical action recognition we also consider their ancestor classes $\mathcal{H} = \{|\mathcal{A}| + 1, |\mathcal{A}| + 2, ..., |\mathcal{A}| + |\mathcal{H}|\}$, which allows us to construct a hierarchical tree with three levels, i.e., grandparent, parent, and child (see Figure 3 right). This process of embedding the hierarchies is performed once, offline, per dataset. However, this process can easily be repeated for alternative hierarchies. **Learning Action Embeddings.** We map the action hierarchy $\mathcal{A} \cup \mathcal{H}$ into the shared hyperbolic space $\mathbb{D}^n$ to obtain hierarchical action embeddings, which are used as action class templates in the next section. Let $\mathcal{P} = \{(u, v) | u = \phi(v)\}$ be the positive pair of $v$ and its parent $\phi(v)$ and $\mathcal{N} = \{(u', v') | u' \neq \phi(v')\}$ be the negative pairs. The discriminative loss akin to Long et al. (2020) is:

$$\mathcal{L}(\mathcal{P}, \mathcal{N}, \mathbf{\Phi}) = \mathcal{L}_H(\mathcal{P}, \mathcal{N}) + \lambda \cdot \mathcal{L}_S(\mathbf{\Phi}), \tag{1}$$

where $\mathbf{\Phi}$ stands for class templates matrix and its $c - th$ column $\mathbf{\Phi}_c$ is the template vector of class $c$ in $\mathbb{D}^n$. For the loss function, $\mathcal{L}_H$ encourages the preservation of parent-child relations and $\mathcal{L}_S$ enforces separation among different sub-hierarchies. The first part $\mathcal{L}_H$ is akin to Nickel & Kiela (2017a), where the wrongly positioned child-parent pairs will be penalized:

$$\mathcal{L}_H(\mathcal{P}, \mathcal{N}) = \sum_{(\boldsymbol{u}, \boldsymbol{v}) \in \mathcal{P}} \log \left( \frac{e^{-d(\boldsymbol{u}, \boldsymbol{v})}}{\sum_{(\boldsymbol{u}, \boldsymbol{v}') \in \mathcal{N}} e^{-d(\boldsymbol{u}, \boldsymbol{v}')}} \right), \tag{2}$$

where $-d(\boldsymbol{u}, \boldsymbol{v})$ is the hyperbolic distance between two action embeddings $\boldsymbol{u}$ and $\boldsymbol{v}$, which can be written in short-hand notation:

$$d(\boldsymbol{v}, \boldsymbol{u}) := 2 \operatorname{arctanh}\left( \| -\boldsymbol{v} \oplus \boldsymbol{u} \| \right), \tag{3}$$

where $\oplus$ indicates the Möbius addition Ungar (2007) in $1-$curved hyperbolic space $\mathbb{D}^n$.

In the second part, we encourage the separation among sibling relationships, where we update $\boldsymbol{\Phi}$ with separation loss:

$$\mathcal{L}_S(\boldsymbol{\Phi}) = - \sum_{i \in |\mathcal{A}|} ||\tilde{\boldsymbol{\Phi}}_i^T \tilde{\boldsymbol{\Phi}}_i||_F + \gamma ||(\hat{\boldsymbol{\Phi}}_i \hat{\boldsymbol{\Phi}}_i^T - \mathbf{I})||_F, \tag{4}$$

where $\hat{\boldsymbol{\Phi}}$ contains non-sibling vectors for action class $i$, while $\tilde{\boldsymbol{\Phi}}$ comprises its sibling vectors.

After learning with the above objectives, we obtain $\boldsymbol{\Phi}$, a matrix of action template vectors including both actions $\mathcal{A}$ and ancestor (parent and grandparent) actions $\mathcal{H}$.

## 3.2 Hierarchical Prototype Explainer

Figure 3 gives an overview of our proposed Hierarchical prototype explainer (HIPE) for video action recognition. HIPE consists of a 3D-CNN backbone $f$ for extracting features from the video frames, and a hierarchical prototype layer $g_p$ for learning prototypes for each frame. The prototype layer is followed by a fully connected layer $h$ that combines the prototype similarity scores and maps them to the shared hyperbolic space through exponential mapping. Prior knowledge about the relations between actions, in the form of the action hierarchy, are projected to the shared space through discriminative embeddings. Subsequently, we use hyperbolic learning to obtain hierarchical prototypes that enable multi-level explainability.

As the backbone architecture, we use the video action classification network 3D-Resnet Hara et al. (2018). For each input video $v \in \mathbb{R}^{W \times H \times T \times 3}$ with $T$ frames it extracts video features $Z \in \mathbb{R}^{W_0 \times H_0 \times T_0 \times D}$ with the spatial resolution $W_0 \times H_0$, frames $T_0$ and channels $D$. A key aspect of this backbone is that $T_0 < T$ due to temporal pooling, as such the features $Z$ are extracted for segments rather than individual frames. Because of the temporal pooling, the prototypes learned by HIPE are spatio-temporal thereby explaining which parts of the segment are indicative of the action in the video.

## 3.3 Hierarchical Prototype Layer

Given the features extracted from the 3D-Resnet $Z$, two layers of $1 \times 1 \times 1$ convolutions with the LeakyReLU activations are added for adjusting the number of channels for the top layer. Hierarchical prototypes are learned by incorporating the prior hierarchical knowledge about actions into the network by representing them as embeddings. We optimize hierarchical prototypes by aligning them to the action embeddings in hyperbolic space.

For each child $\mathcal{A}$ and its parent $\mathcal{H}$ action, the network learns $m$ and $n$ prototypes respectively $P = \{p_j\}_{j=1}^{m+n}$, whose shape is $W_1 \times H_1 \times T_1 \times D$ with $W_1 \leq W_0$, $H_1 \leq H_0$ and $T_1 \leq T_0$. Similar to Chen et al. (2019) we use $W_1 = H_1 = 1$, and in our case we add $T_1 = 1$ for the additional temporal dimension. As such each prototype represents a spatio-temporal part of the video. Given the convolutional output $Z = f(v)$ and prototypes $p$, a prototype layer $g_p$ computes distances between each prototype $p_j$ and patches from $Z$ and converts them to similarity scores using

$$g_p(p_j, Z) = \max_{z \in Z} \log \frac{(||z - p_j||_2^2 + 1)}{(||z - p_j||_2^2 + \epsilon)}, \epsilon > 0 \tag{5}$$

The distances between each prototype and the patch determine the extent to which a prototype is present in the input. We expect to learn different prototypes for child, parent and its grandparent e.g. in Figure 3 the prototype learned for the *applying eye makeup* child class is an eye, for the parent it is focusing on

hairs because of the parent class *self grooming* and shows hand for the grandparent *human object interaction* class. We then multiply similarity scores with the weights of a fully connected layer $h$ to obtain embeddings to be projected in the hyperbolic joint space for learning hierarchical prototypes.

### 3.4 Hierarchical Video Embeddings

The embeddings $\boldsymbol{h} = h(g_p(p, f(v)))$ obtained from the fully connected layer are in the Euclidean space and can not be directly mapped into the hyperbolic embedding space, therefore, we use exponential mapping Ganea et al. (2018) to map video embeddings into the hyperbolic space.

$$\exp_{\boldsymbol{x}}(\boldsymbol{h}) = \boldsymbol{x} \oplus \left( \tanh\left( \frac{||\boldsymbol{h}||}{1 - ||\boldsymbol{x}||^2} \right) \frac{\boldsymbol{h}}{||\boldsymbol{h}||} \right) \tag{6}$$

where $\oplus$ indicates the $1-$curved Mobius addition, $\boldsymbol{x}$ is the tangent point connecting tangent space $\mathcal{T}_0\mathbb{D}^n$ to $\mathbb{D}^n$. Different values of $x$ lead to different tangent spaces, to avoid any ambiguities we set $\boldsymbol{x} = \boldsymbol{0}$ and project the video embeddings to the hyperbolic space for matching with the hierarchical actions.

### 3.5 Training

Our training process consists of a multi-step procedure: In the initial epochs we perform warm-up of the newly added layers. Following the warm-up, we train the entire network end-to-end. Every 10 epochs we update the prototype layer only, followed by a phase of fine-tuning the layers after the prototype layer.

**Video and Action Matching in the Hyperbolic Space**. We aim to learn a latent space where patches important for classification are clustered around similar prototypes. In order to learn hierarchical prototypes we optimize the prototypes $P = \{p_j\}_{j=1}^{m+n}$ to match videos to hyperbolic action embeddings, hence our optimization is supervised by $\boldsymbol{\Phi} \in \mathbb{D}^{n \times (|\mathcal{A}|)}$. Let $\{(v_i, y_i)\}_{i=1}^N$ be the training set, where $v \in \mathbb{R}^{W \times H \times T \times 3}$ and $y_i \in \mathcal{A}$. Our goal is to solve:

$$\mathcal{L}_{crs} + \lambda_1 \mathcal{L}_{cls} + \lambda_2 \mathcal{L}_{sep} \tag{7}$$

**Hierarchical Cross Entropy**. The first term in our loss is the hierarchical cross-entropy loss $\mathcal{L}_{crs}$ which penalizes the misclassification, and is defined as:

$$\mathcal{L}_{crs} = \frac{1}{N} \sum_{i=1}^{N} \sum_{k=1}^{K} y_{ik} \log p(y = k | v) \tag{8}$$

The Softmax in the cross entropy is defined as the negative distance between video embeddings and the hierarchical action embeddings in the hyperbolic space:

$$p(y = k | v) = \frac{\exp(-d(\boldsymbol{h}_e), \boldsymbol{\Phi}_k)}{\sum_{k'} \exp(-d(\boldsymbol{h}_e), \boldsymbol{\Phi}_k))}, \tag{9}$$

where $\boldsymbol{h}_e = \exp_{\boldsymbol{0}}(\boldsymbol{h})$ is applying exponential map to the fully connected layer output $\boldsymbol{h}$.

**Hierarchical Clustering**. In order to provide meaningful explanations at each level of hierarchy our hierarchical clustering cost encourages input frames to have at least one patch from features to be closer to a child, parent or grandparent class prototype.

$$\mathcal{L}_{cls} = \frac{1}{N} \sum_{i=1}^{N} \min_{j:p_j \in P_{|\mathcal{A}|+|\mathcal{H}|}} \min_{z \in patches(f(v_i))} ||z - p_j||_2^2 \tag{10}$$

**Hierarchical Separation**. Our hierarchical separation cost encourages the latent patches of the frames to stay away from the prototypes not belonging to the same child class or parent class or grandparent class.

$$\mathcal{L}_{sep} = -\frac{1}{N} \sum_{i=1}^{N} \min_{j:p_j \notin P_{|\mathcal{A}|+|\mathcal{H}|}} \min_{z \in patches(f(v_i))} ||z - p_j||_2^2 \tag{11}$$

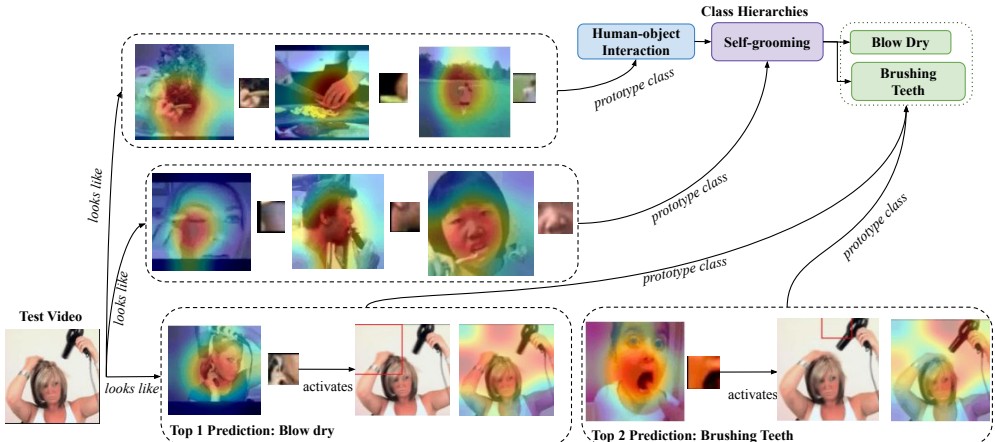

Figure 4: **Hierarchical Explanations.** This example shows the prototypes from grandparent *human-object interaction* class, parent *self-grooming* class and ground truth *blow dry* class, we also observe that the top 2 prediction for the model is its sibling *brushing teeth* class. This conforms that our model is learning hierarchical relations between classes.

## 3.6 Updating Hierarchical Prototype Layer

We project prototypes onto the closest video features from the training videos. We do so for child, parent, and grandparent action categories. Mathematically, for the prototypes $p_j$ from child, parent and grandparent class i.e. $p_j \in P_{|\mathcal{A}|+|\mathcal{H}|}$, we update the prototype layer as:

$$p_j \leftarrow \underset{z \in \mathcal{Z}_j}{\mathrm{argmin}}||z - p_j||_2 \tag{12}$$

where $\mathcal{Z}_j = \{\tilde{z} : \tilde{z} \in patches(f(v_i)) \, \forall i \, \text{s.t.} \, y_i = |\mathcal{A}| + |\mathcal{H}|\}$. Our prototype layer is updated not only with the prototypes belonging to the child class but also with the parent and grand parent classes enabling the learning of hierarchical relations between classes.

## 3.7 Hierarchical Prototype Visualization

To construct the visualizations the learned prototypes are mapped to the spatio-temporal input space. We select the patch which highly activates for the prototype $p_j$ by forwarding the input $v$ through the network and upsampling the activation map generated by the prototype layer $g_p(p_j, Z)$ both spatially and temporally (for videos). We visualize $p_j$ for child, parent, and grandparent classes providing explanations at all levels.

# 4 Experimental Setup

## 4.1 Datasets

To evaluate HIPE for videos we conduct experiments on two video datasets: UCF-101 Soomro et al. (2012) and Activity-Net1.3 Caba Heilbron et al. (2015).

**Hierarchical UCF-101.** UCF-101 Soomro et al. (2012) contains 13,320 videos belonging to 101 action classes with a total length of 27 hours. We define one additional level of hierarchy with the number of classes at level one, two, and three being 5, 20, and 101 respectively. The classes at the third level of the hierarchy are the 101 original classes of the dataset. The complete hierarchy is included in the appendix.

**Hierarchical ActivityNet.** ActivityNet Caba Heilbron et al. (2015) contains 14,950 untrimmed videos with each video consisting of one or more action segments belonging to 200 action classes with a total length

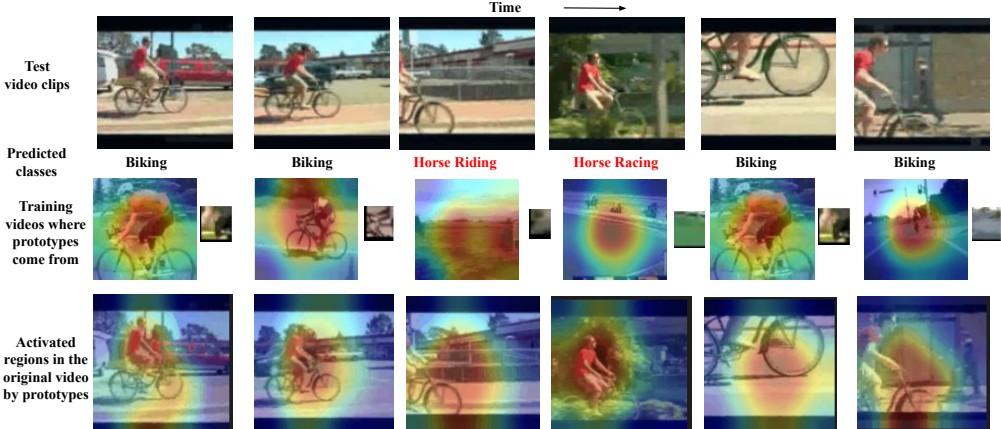

Figure 5: **Spatio-temporal Explanations.** Explanations for a video with six clips. Top row: one frame per clip. Second row: class predictions. Third row: training video frames where prototypes came from. Bottom row: activated regions in the original video frames. When the network is focusing on more abstract concepts like grass and barrier bars it gets misclassified into *horse riding* class. Similarly, when it is focusing on grass in the field it gets misclassified into *horse racing* class. While when it focuses on more concrete concepts like bicycle tyres or handle it is correctly classified.

of approximately 648 hours. We use 10,024 videos for training and 4,926 for validation. We follow Hara et al. (2018) and train and test our model on trimmed videos to determine video-level accuracy. We follow Long et al. (2020) and use the class-level action hierarchies defined by them. It contains 200, 38, and 6 classes in level one, two, and three respectively.

## 4.2 Implementation Details

The hierarchical action embeddings are generated by training the model with Reimannian Adam optimizer Ganea & Bécigneul (2018), implemented with *geoopt* and Pytorch Paszke et al. (2019). Apart from the one-time offline step of generating the hierarchical action embedding HIPE is trained in an end-to-end fashion. For feature extraction, we used Resnet-3D-18 Hara et al. (2018) pre-trained on Kinetics Carreira & Zisserman (2017) and added two $1 \times 1$ convolutional layers with the LeakyReLU, a prototype layer and the final embedding layer. We perform prototype projection and visualization every 10 epochs.

We report results on two variations of HIPE: we compare between prototype projection with 5 prototypes per class and 10 prototypes per class, to explore whether this additional supervision makes it possible to use fewer prototypes. For comparison, we adapt ProtoPNet Chen et al. (2019) to videos by replacing the 2D ResNet backbone with a 3D ResNet.

**Evaluation Metrics.** We report the performance at both clip level and video level. Additionally, to show the benefit of using hierarchical learning, we report accuracy for three metrics: the class accuracy is calculated as the rate of correct prediction in the hierarchical space 0-hop away from the ground-truth, the sibling accuracy as the rate of correct prediction 2-hops away from the ground-truth, and the cousin accuracy as the 4-hops correct prediction rate. Higher performance on the sibling and cousin metrics indicates that misclassifications are to hierarchically nearby, and therefore, semantically similar classes.

## 5 Visual Explanations

**Hierarchical Explanations.** Figure 4 shows multi-level explanations provided by HIPE. Our model learns to represent the video as hierarchical prototypes that belong to grandparent, parent and child classes. For example, in Figure 4 our model has learned prototypes (only three out of ten prototypes shown for better presentation) for the grandparent class *human-object interaction*, parent class *self-grooming*, and the action

| | Network | Accuracy ($\sigma$) | Sibling Accuracy | Cousin Accuracy | # of prototypes per class |
|---|---|---|---|---|---|
| Non-Interpretable Models | 3D-Resnet | 83.54 (0.21) | 89.73 | 93.62 | - |
| | Resnet-Hyperbolic | 81.64 (1.22) | 89.99 | 93.28 | - |
| Interpretable Models | ProtoPNet | 77.89 (0.50) | 85.92 | 90.98 | 10 |
| | TesNet | 75.39 ($--$) | 79.44 | 85.03 | 10 |
| | Euclidean HIPE | 77.18 ($--$) | 86.72 | 91.60 | 10 |
| | HIPE | 79.45 ($--$) | 88.88 | 92.73 | 5 |
| | HIPE | **80.57** (0.24) | **89.30** | **93.02** | 10 |

Table 1: **Clip level accuracy comparison for different models on UCF-101 videos.** We observe that HIPE with 10 prototypes per class recovers the drop due to accuracy-explainability trade off significantly while providing multi-level explanations. Accuracy is reported as the mean of three runs for each model with the standard deviation.

| | Network | Accuracy | Sibling Accuracy | Cousin Accuracy | # of prototypes per class |
|---|---|---|---|---|---|
| Non-Interpretable Models | 3D-Resnet | 49.99 | 51.59 | 63.88 | - |
| | Hyperbolic-Resnet | 49.95 | 52.03 | 65.44 | - |
| Interpretable Models | ProtoPNet | 46.06 | 47.74 | 61.67 | 10 |
| | HIPE | 45.67 | 48.72 | 62.84 | 5 |
| | HIPE | **46.26** | **48.93** | **62.97** | 10 |

Table 2: **Clip level accuracy comparison for different models on ActivityNet videos.** We observe that HIPE with 10 prototypes per class recovers the drop for siblings and cousins and shows comparable performance with regular ProtoPNet for class accuracy while providing multi-level explanations.

class *blow dry* (only one prototype and its activation on the original video shown). We also observe that the second most likely prediction is its sibling class *brushing teeth*.

**Spatio-temporal Explanations.** Figure 5 shows explanations for a video with six clips of 16 frames. The top row shows single frames for each clip. In the second row we observe that the model fails to predict correct classes for third and fourth clip. The bottom row shows training video frames where prototypes came from (one prototype per clip shown for better visibility) and classification is based on. We see that when the network is focusing on the more concrete concepts like the tyres of the bicycle, or handle it gets correctly classified. While when the network is focusing on the grass and barrier bars, and it activates bars in the background of the cyclist in original clip, it gets misclassified into the *horse riding class*. Similarly, when it is focusing on grass in the field, it activates greenery in the background of the cyclist in original clip, it gets misclassified into *horse racing class*. The parent and grandparent classes for all the clips are still the same i.e., *riding sports* and *sports* respectively, giving us useful information even in case of misclassification.

# 6 Accuracy-Explainability Trade Off

In order to explore the accuracy-explainability trade-off for HIPE we present quantitative comparison of our model against interpretable and non-interpretable baselines.

**Non-Interpretable Models.** The performance of non-interpretable models on UCF-101 and ActivityNet are shown in the top two rows of Tables 1, 2, and 3. For fair comparison both non-interpretable models, a regular Resnet Hara et al. (2018) and a hyperbolic Resnet Long et al. (2020), are trained end-to-end with the same data augmentations and an equal number of epochs. The only difference between the two non-interpretable models is that for the Resnet model the categories are separated through euclidean hyperplanes while the Resnet-Hyperbolic utilizes hyperbolic embedding space to separate categories. Our results for UCF-101 show that both a regular Resnet and the hyperbolic Resnet perform similarly at clip level (Table 1) and video level (Table 3).

|  | Network | UCF-101 Accuracy | ActivityNet Accuracy | # of prototypes per class |
|---|---|---|---|---|
| Non-Interpretable Models | 3D-Resnet | 87.92 | 69.15 | - |
|  | Resnet-Hyperbolic | 87.49 | 70.19 | - |
| Interpretable Models | ProtoPNet | 84.48 | 66.46 | 10 |
|  | HIPE | 86.25 | 63.82 | 5 |
|  | HIPE | **87.07** | **66.48** | 10 |

Table 3: **Video level accuracy comparison for different models on UCF-101 and ActivityNet videos.** We observe that HIPE with 10 prototypes per class recovers the drop at video level significantly for UCF-101 and shows comparable performance with the regular ProtoPNet for ActivityNet while providing multi-level explanations.

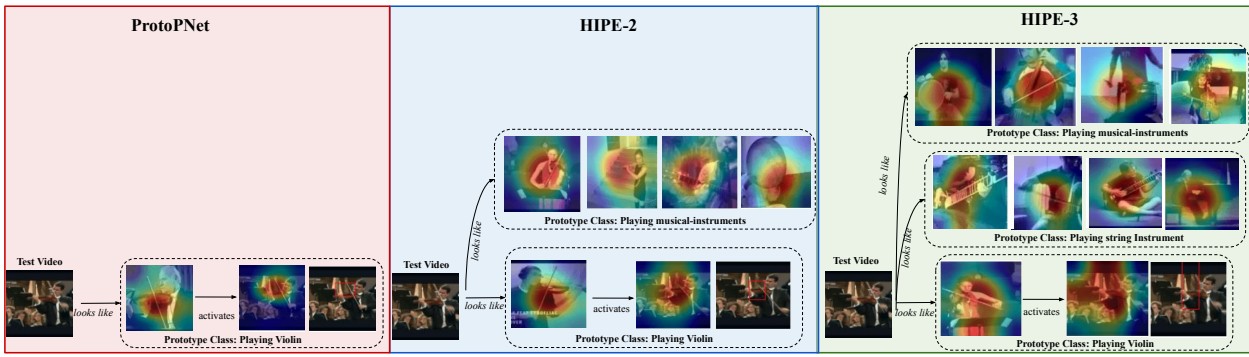

Figure 6: **Significance of Hierarchy for Explanations.** ProtoPNet provides explanations at a single level (*playing violin*). On the other hand, HIPE-2 offers explanations not only at the children level (*playing violin*), but also at a more abstract level (*playing musical instruments*). Moreover, HIPE-3 provides explanations at the children level (*playing violin*), the parent level (*playing string instrument*), and even the grandparent level (*playing musical instruments*). Note that HIPE-3 and HIPE are the same, here called HIPE-3 for clarity.

For ActivityNet the clip-level class accuracy (Table 2) is comparable across both networks, however, due to the hierarchical learning of hyperbolic Resnet it shows better sibling and cousin accuracy, additionally, it shows an improvement for class accuracy at the video level (Table 3). Overall, we see comparable performance for the non-interpretable networks on UCF-101 and improvements with Hyperbolic networks for ActivityNet.

**Interpretable Models.** The performance of interpretable models on UCF-101 and ActivityNet are shown in the bottom three rows of Table 1, 2, and 3. We report the results for a regular ProtoPNet Chen et al. (2019) and TesNet Wang et al. (2021) adapted for videos and the variations of HIPE euclidean HIPE, HIPE with 5 and 10 prototypes. For UCF-101, with a regular ProtoPNet with 10 prototypes per class, the accuracy drops considerably: the clip-level class accuracy drops to 77.89 and the video-level accuracy to 84.48. This is because of the explainability-accuracy trade off common in explainable-AI, also reported in Chen et al. (2019). For TesNet the performance is 2.50 below ProtoPNet - we suspect this is due to the mapping of the features to the Grassmann manifold not functioning well in the 3D case. Euclidean HIPE maps hierarchical embeddings to Euclidean space instead of hyperbolic space. Although clip-level class accuracy remains comparable to ProtoPNet, sibling and cousin accuracy improve due to the inclusion of hierarchical information, despite operating in Euclidean space. In contrast, HIPE with 5 prototypes per class is much less affected and recovers the drop by 1.56 for class accuracy, 2.96, and 1.75 for sibling and cousin accuracies respectively. Increasing the number of prototypes to 10 per class further improves the performance by 2.68, 3.38, and 2.04 for class, sibling, and cousin accuracies respectively. Moreover, the performance at the video level reaches 86.25 (see Table 3). Hence, both variations of HIPE reduce the accuracy drop.

On ActivityNet (see Table 2) we observe a clear accuracy-explainability trade off for the regular ProtoPNet, with drops in both the clip and video level accuracies. However, whilst HIPE shows a similar drop in class

| Network | Accuracy | Sibling Accuracy | Cousin Accuracy |
|---|---|---|---|
| HIPE-2 (two levels of hierarchy) | **82.04** | **93.02** | - |
| HIPE-3 (three levels of hierarchy) | 80.40 | 89.30 | 93.02 |

Table 4: **Significance of hierarchies for accuracy-explainability trade-off on UCF-101 videos.** HIPE-2 shows better accuracy than HIPE-3 , however HIPE-2 provides explanations at two levels and HIPE-3 at three levels of granularity. Note HIPE-3 and HIPE are the same, here called HIPE-3 for clarity.

| Model | Accuracy |
|---|---|
| With Non-hierarchical Clust & Sep | 79.49 |
| With Hierarchical Clust & Sep | **80.40** |
| Without Hierarchical Sep | 80.28 |
| Without Hierarchical Clust | - |
| Without Clust & Sep | 77.19 |

Table 5: **Significance of separation and clustering loss on UCF-101.** Without hierarchical clustering leads to unstable model.

accuracy we can observe that it partially recovers from this drop on the sibling and cousin metrics. This behavior holds for both the HIPE with 5 prototypes, and for the variant with 10 prototypes per class, we even see improvements for the sibling and cousin metrics of 1.19 and 1.30 respectively. Whilst ActivityNet remains challenging, an improvement in sibling and cousin accuracies is directly beneficial to the explainability as demonstrated in Section 5.

Hence, we can observe that on both datasets HIPE is less affected by the accuracy-explainability trade off whilst also providing multi-level explanations.

**Significance of Hierarchy.** Table 4 compares the performance of HIPE with two levels of hierarchy (HIPE-2) and three levels of hierarchy (HIPE-3). We observe that HIPE-2 further improves the performance compared to HIPE-3. However, it is important to note that the explanations provided by HIPE-2 are limited to two levels, indicating the presence of an accuracy-explainability trade-off.

Figure 6 illustrates the significance of hierarchy in explanations. ProtoPNet provides explanations at a single level (*playing violin*). In contrast, HIPE-2 offers explanations not only at the children level (*playing violin*), but also at a more abstract level (*playing musical instruments*). Additionally, HIPE-3 provides explanations at the children level (*playing violin*), the parent level (*playing string instrument*), and even the grandparent level (*playing musical instruments*).

**Significance of Separation and Clustering loss.** Table 5 compares the performance of HIPE with and without the hierarchical clustering and separation losses. We observe that introducing hierarchical clustering and separation enhances the performance. Removing both of them leads to around 3% drop in the performance. Using only separation without clustering destabilises training, while clustering without separation performs worse than using both.

## 7 Conclusion

In this work, we proposed Hierarchical prototype explainer for video action recognition. By learning hierarchical prototypes we are able to provide explanations at multiple levels of granularity, not only explaining why it is classified as a certain class, but also what spatiotemporal parts contribute to it belonging to parent categories. Our results show that HIPE outperforms a prior non-hierarchical approach on UCF-101, whilst performing equally well on ActivityNet. Additionally, we demonstrate our multi-level explanations that make it possible to see which spatiotemporal parts contribute to grandparent, parent, and class-level classifications. Our hierarchical approach thereby provides richer explanations whilst compromising less performance to gain explainability.

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

## A  Appendix

### A.1  Hierarchies for UCF-101 and ActivityNet

Figure 7 and 8 show the hierarchies for UCF-101 Soomro et al. (2012) and ActivityNet Caba Heilbron et al. (2015) respectively.

Two levels of hierarchy are given with UCF-101 we define one extra level with the number of classes at level one, two, and three being 5, 20, and 101 respectively. For example, one of the grand parent class is *playing music*, parents are *wind, string, percussion* and children are *playing flute, playing guitar, drumming* and more. The classes at the third level (i.e., child level) of the hierarchy are the 101 original classes of the dataset.

ActivityNet contains 200, 38, and 6 classes in level one, two, and three respectively (see Figure 8). For instance, one of the grand parent is *personal care* and the parents are *dress up, grooming, wash up* and children are *putting on shoes, getting a haircut, shaving* etc.

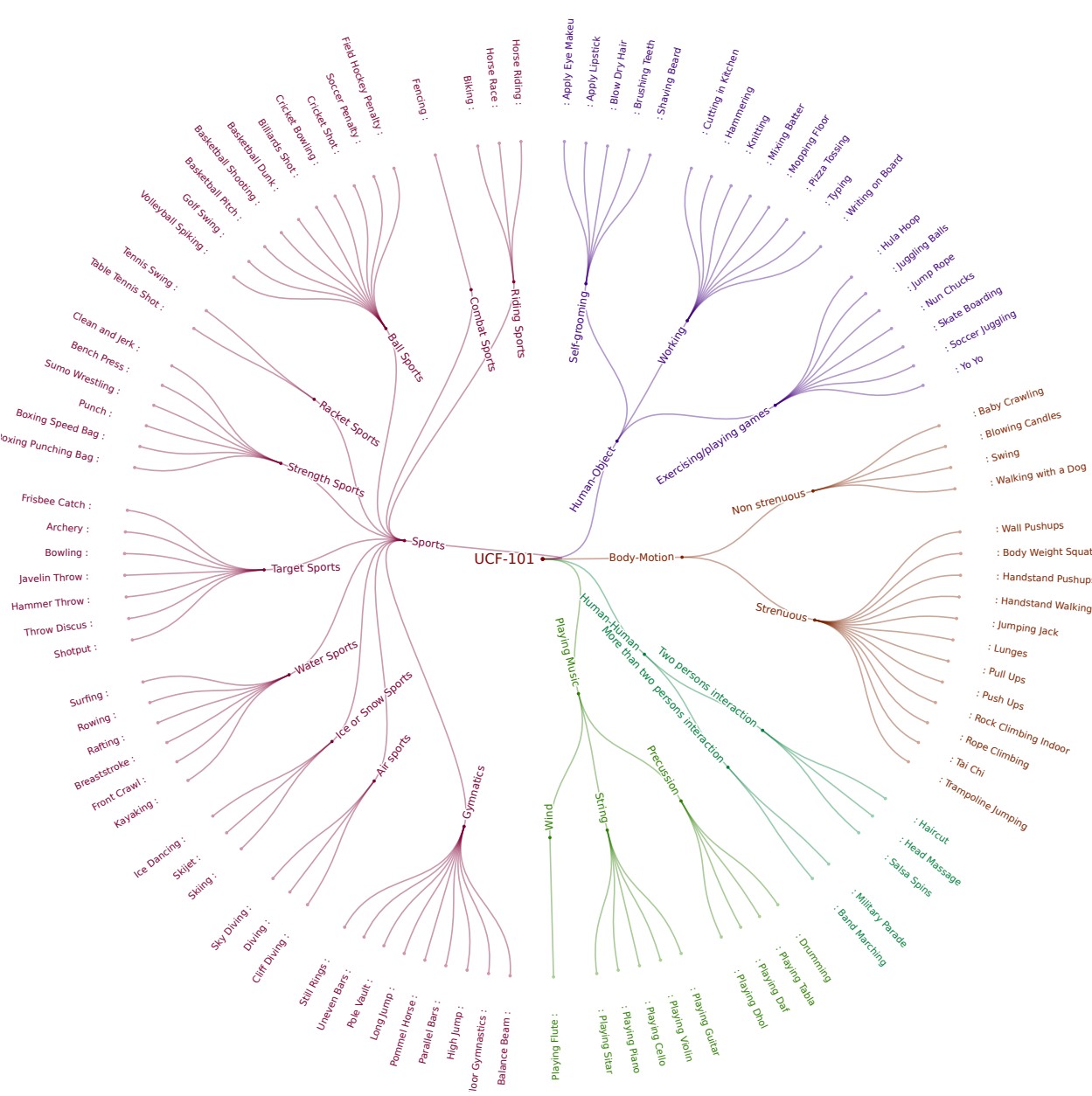

Figure 7: **Hierarchy for UCF-101.** Schematic representation of the hierarchy defined for UCF-101 dataset. The three levels of hierarchy are grand parent (*playing music*), parent (*wind, string, percussion*) and children (*playing flute, playing guitar, drumming* etc) classes.

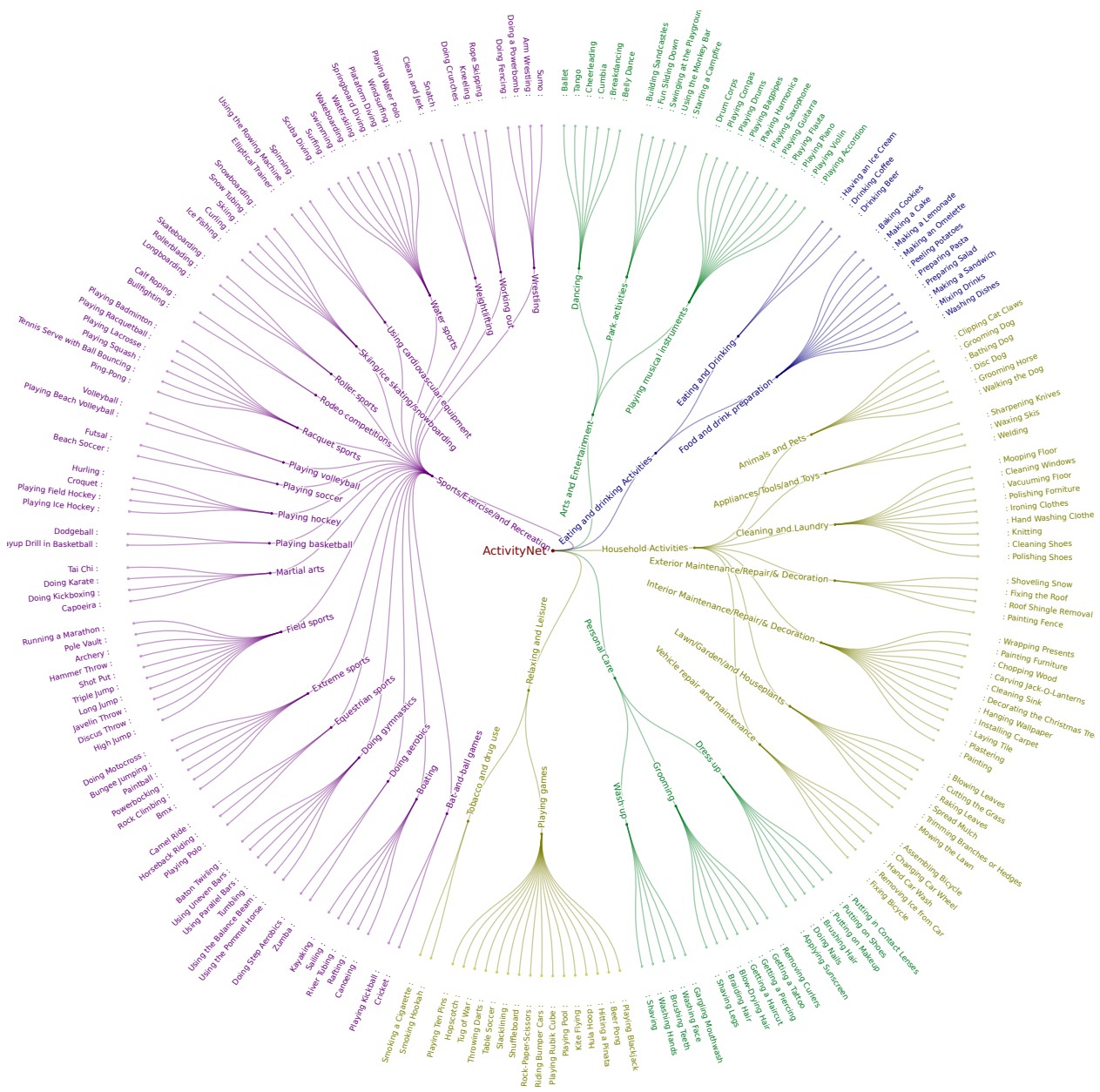

Figure 8: **Hierarchy for ActivityNet.** Schematic representation of the hierarchy defined for Activity dataset. The three levels of hierarchy are grand parent (*personal care*), parent (*dress up, grooming, wash up*) and children (*putting on shoes, getting a haircut, shaving* etc) classes.

## A.2 Increasing the Number of Prototypes

To explore the influence of the number of prototypes on the performance we show additional experiments in Table 6. These results show that the performance saturates beyond 10 prototypes per class, with a small decrease at 15 prototypes. Moreover, for explainability we argue that it is beneficial to have a lower number of prototypes. This finding of 10 prototypes appears to be in line with Chen et al. (2019), who similarly found 10 prototypes per class to work well.

| Network | Accuracy | Sibling Accuracy | Cousin Accuracy | # of prototypes per class |
|---------|----------|------------------|-----------------|---------------------------|
| HIPE | 79.45 | 88.88 | 92.73 | 5 |
| HIPE | **80.57** | **89.30** | **93.02** | 10 |
| HIPE | 80.17 | 88.81 | 92.77 | 15 |

Table 6: **Clip level accuracy for varying numbers of prototypes per class on UCF-101 videos.** We observe that HIPE with 10 prototypes per class performs best, and that performance degrades when increasing or decreasing the number of prototypes.

