# OpenReview forum: "Hierarchical Prototype-based Explanations"
_TMLR — Rejected by TMLR_

### Review · Reviewer_supD · 2024-04-12

**Summary Of Contributions:**

The authors introduce the Hierarchical Prototype Explainer (HIPE), an approach for interpreting the actions recognized by deep neural networks in video data. This method stands out by leveraging hierarchical relationships between action classes, offering multi-level explanations that mirror human reasoning processes. The authors demonstrate HIPE's effectiveness on two video datasets, ActivityNet and UCF-101.

**Audience:**

Yes

**Claims And Evidence:**

Yes

**Requested Changes:**

Please try to address the weaknesses mentioned above.

**Strengths And Weaknesses:**

Strengths:
- The introduction of hierarchical prototype-based explanations for video action recognition represents a novel advancement in the field. By mimicking human cognitive processes, HIPE addresses the complexity of interpreting actions in videos, providing multi-level explanations that enhance understandability.
- The paper successfully demonstrates that HIPE can reduce the often observed trade-off between model accuracy and explainability.
- The use of hyperbolic spaces to organize hierarchical knowledge is both innovative and effective.

Weaknesses:
- While the paper presents results on two datasets, the generalizability of HIPE to other datasets or domains remains to be fully explored. Further testing across a wider range of video data, including those with more complex or subtle hierarchical relationships, would be valuable.
-  A more detailed discussion of how the number of prototypes affects both performance and explainability would help guide practical implementations.
- The work introduces an innovative approach to utilizing hierarchical levels for explanations but could further discuss the impact of varying the depth and breadth of the hierarchy on model performance and explainability.

---

> ### Author Response · Authors · 2024-05-09
> **Review Response**
>
> # Response
>
> We thank the reviewer for their positive comments on our contributions to hierarchical explanations for videos and reducing the accuracy-explainability trade-off, as well as for recognising our innovative and effective use of hyperbolic spaces. Below we further address the points brought up for change.
>
> ## Additional datasets
> We agree with the reviewer that this would be valuable, and will publish the code on publication of the paper to enable the community to explore our method on other datasets. Within the scope of the paper we do believe that the two datasets studied have sufficient variety to support our claims.
>
> ## Influence of number of prototypes
> We thank the reviewer for this suggestion. In the paper we compare between using 5 and 10 prototypes and found 10 prototypes to work better, to further explore the influence of the number of prototypes we perform an additional experiment with 15 prototypes on UFC101 (reported in Appendix A.2), and show the results here for convenience.
>
> | Network | Accuracy | Sibling Accuracy | Cousin Accuracy | # of prototypes per class |
> | -------- | -------- | -------- | -------- | -------- |
> | HIPE  | 79.45  | 88.88  | 92.73  | 5 |
> | HIPE  | **80.57**  | **89.30**  | **93.02**  | 10 |
> | HIPE  | 80.17  | 88.81  | 92.77  | 15 |
>
>
> These results confirm that 10 prototypes per class works best, which is in line with what the ProtoPNet paper found. We do note that this may be dataset specific to some extent, however, 10 per class appears to be a solid default. In terms of explainability we see an inverse relationship, where more prototypes reduces explainability due to information overload. We have included this discussion in the Appendix A.2.
>
> ## Hierarchy depth and breadth
> We thank the reviewer for bringing up this discussion on the depth and breadth of hierarchy. A challenge for this discussion is that the definition of hierarchy is highly subjective and limited by the semantics of the dataset. Therefore, we can only do limited discussion based on the datasets studied.
>
> We explore depth, the difference between 2-layer and 3-layer hierarchies, in Table 4, and actually find that a shallower hierarchy increases the performance. For breadth, we may regard the prototypes of each category as leaf nodes of the hierarchy, and then the number of different prototypes can also be regarded as the breadth of the hierarchy, as shown in Table 3 and appendix A.2. For this interpretation of breadth we see some effect, as discussed above.
>
> Addressing breadth as in varying the number of low-level concepts is similarly challenging, as this alters the dataset semantics. In comparing UFC101 and and ActivityNet, with 101 and 200 low-level concepts respectively, we observe that all methods obtain lower accuracy scores on the latter dataset. Hence, we cannot extrapolate these findings to the hierachy breadth.
>
> Both sets of discussions are limited to the hierarchy definition of the dataset itself. Since the dataset itself does not come with other granular hierarchy definition methods, it is not feasible to further discuss its depth and breadth impact.

---

### Review · Reviewer_CmWu · 2024-04-18

**Summary Of Contributions:**

This work aims to provide explanations for deep neural networks used for video action recognition. Based on the idea of matching prototypes in a hierarchical way, the paper proposed a novel framework that first learns prior hierarchical action embeddings in a hyperbolic space and then clusters video frames of actions around similar prototypes. Experimental results show that the proposed method can achieve a better accuracy-explainability trade-off and explain the reason for the model’s predictions for video actions.

**Audience:**

Yes

**Broader Impact Concerns:**

I have no concerns about the ethical implications of the work.

**Claims And Evidence:**

Yes

**Requested Changes:**

See the weaknesses stated above.

Minor: Please add the date for the citation “Oikarinen et al.”

**Strengths And Weaknesses:**

Strengths

- The paper comprehensively discusses the related works that cover relevant approaches and clearly situated the proposed approach in the literature.

- The methodology is clearly explained with nice figures.

- Experimental results conducted on two different datasets show the proposed method is superior than previous ones.


Weaknesses

- I think the motivation for this work can be stated more clearly. I am uncertain about the core target issue this paper tries to solve: is it dealing with uncertainty (abstract), providing descriptions at different levels (1st paragraph of intro), exempting from the need for dense annotations (2nd paragraph), achieving better accuracy-explainability trade-off (4th paragraph), making the explanation more understandable by everyone (1st paragraph of Section 3). The proposed method deals with all of them at some level, but I believe the paper would benefit from a more organised introduction and consistent motivation.

- It’s hard to fully understand the usefulness of the proposed method. It is less accurate than the non-interpretable models anyway. In what situations can the users benefit from the hierarchical explanations?

- Learning embeddings in the hyperbolic space is claimed to be an important aspect of the proposed method, yet there is no ablation study on this idea. How will the proposed method work in the Euclidean space?

---

> ### Author Response · Authors · 2024-05-09
> **Review Response**
>
> # Response
>
> We thank the reviewer for their positive comments on our related work and method description. Below we further address the points brought up for change. In addition we have fixed the minor point in the revised paper.
>
> ## Ablation
> We thank the reviewer for suggesting the ablation study, we have included the results in Table 1 and summarized them in the shared response above. We implemented HIPE in the euclidean space by keeping the euclidean space, rather than mapping them to hyperbolic space. We observe that the class level accuracy is slightly below ProtoPNet, however the sibling and the cousin accuracy improves because of the added hierarchical information even in euclidean space. We believe these results further validate our method design.
>
> ## Motivation of the paper
> We thank the reviewer for their critical reading of our introduction, and recognise that our explaination was insufficiently consistent. To remedy this, we have in the revised paper (marked in red), updated the wording to  clarity that the main motivation of our work is to provide meaningful explanations in situations where there is uncertainty due to lack of information. In addition, two key benefits of our approach are that we do not require dense annotations and achieve a better accuracy-explainability trade-off.
>
> ## Benefits of hierarchical explanations
> We discuss the accuracy-explainability trade-off in the shared response, however, additionally users may benefit from hierarchical explanations specifically in situations where there are many fine-grained low-level concepts. In Figure 2 we illustrate such a case where a user may not be familiar with the 'daf' instrument, and as such explanations based on this concept may not be useful. However, also offering explanations for 'percussion' may be helpful for such a user.
>
> Additionally, in fine-grained cases there may be many shared visual characteristics between concepts (e.g., kayaking and rafting are highly similar, with the shape of the boat and the number of passengers being the distinction). Hierarchical explanations allow users to zoom in on the concept-specific prototypes for why the classification was made.

---

### Review · Reviewer_bwcW · 2024-04-26

**Summary Of Contributions:**

This paper studies the problem of interpret deep neural networks. Specifically, targeting one type of existing approach that derive prototypes based on dissecting the visual inputs to explain the classification results, hierarchy is introduced upon the prototypes to explain the concepts at different levels.

The motivation of the proposed method is from the observation on how human recognize and understand the videos, which is described in the water sports example in the paper.

The proposed method is tested on two public video datasets, the results indicate that the proposed method perform well within the intepretable models.

**Audience:**

Yes

**Claims And Evidence:**

Yes

**Requested Changes:**

1. Please give more explanation on the example of water sports and human swimming activity given in the abstract, which represents the motivation of the paper but is a little bit confusing.

2. I would recommend tha authors to polish the description of the background and motivation of the target problem, so that readers unfamiliar with this field can more easily understand the work.

3. Please more on why it is important to explain the semantic concepts at both high and low levels. It seems that only the lowest level of concept is necessary, because all the parent levels can be inferred. For example, if the lowest level of concept observed is 'rafting', without other information from the video, we can know the concept of water sports and sports should present.

**Strengths And Weaknesses:**

Strengths:

1. The proposed method HIPE obtains higher accuracy than the interpretable baselines, which demonstrates that the proposed model can strike a better balance between accuracy and interpretability.

2. Unlike the previous works on generating human understandable concepts, this work does not require dense annotation for model training.

Weakness:

1. It is unclear why different levels of concepts should be directly detected from the video. This point is elaborated in 'Requested changes'.

2. The proposed method still performs worse than the non-interpretable methods, which indicates a gap between human-friendly representation and the machine-friendly representation.

---

> ### Author Response · Authors · 2024-05-09
> **Review Response**
>
> # Response
> We thank the reviewer for their positive assessments of our contributions in closing the accuracy-explainability trade-off without the need for dense annotations. Below we further address the points brought up for change.
>
> ## Detecting high-level concepts and motivation of the paper
> As also discussed in the share response, our work is motivated by situations when there is insufficient information to detect a low-level concept, which, for example, may occur in videos due to occlusion or the main subject being out of frame. In such cases non-hierarchical explanations are would have to be grounded in an inaccurate classification, whereas our model can use higher level concepts.
>
> Figure 1 illustrates such a case, where at t1, t2, and t4 we can clearly recognise it as rafting, but at t3 there is insufficient information to recognise it as such. By having prototypes at multiple levels we can ground the explanation in prototypes related to watersports. Based on your and reviewer **CmWu**'s suggestions we have adjusted the introduction to more consistently explain our motivation.
>
> An additional benefit of hierarchical explanations is that more general prototypes applicable to multiple low-level concepts can be shared between concepts, whereas in a non-hierarchical approach the models may need to learn redundant information.
>
> ## Gap between human-friendly and machine-friendly representation
> Indeed, there remains a gap between human-friendly representation and machine-friendly representation, and this is a general challenge within explainability literature. However, a major contribution of our work is that we reduce this gap, and thereby offer a better alternative for situations where explainability is a requirement, but good performance is also still desirable. We further elaborate on this point in the shared response.

---

### Author Response · Authors · 2024-05-09
**Shared Review Response**

# Response

We thank the reviewers for their constructive comments and suggestions. A revised version of the paper has been uploaded, with all changes marked in red. In the response below we address three shared points, and we address further points in the responses to the individual reviews.

## Motivation

In this work we explore incorporating hierarchical information into deep neural networks for video classification for better explainability. By explicitly modelling the hierarchical relations between classes it enables the network to better deal with uncertainty stemming from a lack of information. Particularly in videos, not every frame may contain the necessary information to accurately assign a low-level. In such cases it is beneficial if the network is able to assign a more high-level concept, e.g., in the presence of water all water sports are equally likely - thus explanations based on the presence of water are meaningful in showing a user why a water sport was selected, but less so for explaining the choice between specific water sports.

As pointed out by Reviewer **bwcW**, indeed, if the low-level concept can be detected we can infer the presence of the higher-level concepts. However, our work is motivated by those situations when there is insufficient information to detect a low-level concept, as for these we may use our knowledge of higher-level concepts to still provide meaningful explanations.

Moreover, by allowing the network to share prototypes for higher-level concepts, the prototypes for low-level concepts can focus on the specific details that distinguish them. This allows for better modelling of the hierarchical relations between concepts, which as we show in the paper leads to increased performance over a non-hierarchical approach.

To further clarify our motivation we have adjusted the introduction to emphasise the goal of providing meaningful explanations even when confronted with uncertainty due to lack of information - which we address by incorporating hierarchy.

## Accuracy-Explainability Trade-Off

As noted by Reviewer **supD**, one of the strengths of our approach is that it reduces the often observed trade-off between model accuracy and explainability. This trade-off often leads to a difficult decision, as there are obvious reasons for why explainability may be necessary (e.g., legal reasons like GDPR, model debugging, or for understanding which features are used), yet, we would want to compromise the classification accuracy as little as possible. By reducing this trade-off and reducing the gap between interpretable and non-interpretable models we make this decision easier.

The situations where users may benefit from hierarchical explanations Reviewer **CmWu**, thus include any situation where external factors necessitate model explanability and where the user would like to trade as little performance for explainability as possible. As well as situations where there frequently may be insufficient information to assign a low-level concept, such as for video classification.

## Ablation of Hyperbolic Space

We thank reviewer **CmWu** for the suggestion of an extra ablation of our model without hyperbolic space, and instead learn the hierarchy in euclidean space. We have added these results to Table 1 in the paper, and add them here for your convenience.


| Network | Accuracy | Sibling Accuracy | Cousin Accuracy |
| -------- | -------- | -------- | -------- |
| ProtoPNet     | 77.89     | 85.92     | 90.98 |
| Euclidean HIPE     | 77.18    | 86.72    | 91.60 |
| HIPE     | 80.57     | 89.30     | 93.02 |


From these results we can observe that learning the hierarchy in Hyperbolic space is beneficial for the performance, and that without there is some benefit to the sibling and cousin accuracy, but a slightly decreased class accuracy, thereby underscoring our findings.

---

### Decision · Action_Editor_gThT · 2024-06-14

**Recommendation:** Reject

**Comment:**

After the reviews and rebuttal, reviewers cannot come to a consensus. 2 reviewers are leaning toward acceptance (with 1 quite reluctantly) and 1 reviewer still stays on the rejection side. They still question the motivation of adopting hierarchical explanations. The authors countered that by saying that some of the times the lower level explanation is not available, which is a valid explanation but not shown by any evidence in the paper. Besides, the authors compared against ProtoPNet, but their HIPE also has a difference in terms of using a hyperbolic space instead of a Euclidean space, which is leading to quite significant performance gains.

The Associate Editor read the paper, reviews and author response, and believed that this paper has enough merits, but those merits need to be better illustrated. The authors need to provide:

1) Qualitative examples showing when only higher-level explanations are available and lower-level explanations are not, and that their algorithm is able to bring some advantages in those scenarios. This is in response to that reviewers did not find the motivation to be illustrated satisfactorily by the experiments.
2) Result using their HIPE without a hierarchical explanation, as well as HIPE-2 on cousin accuracy. This is for a more rigorous comparison between hierarchical and non-hierarchical explanations.
3) Potentially some use cases that illustrate that the hierarchical explanation is useful. Qualitative examples showing the hierarchical explanations, prototypes and some user using it to gain better understanding would be useful. A user study would make the paper even stronger.

It is OK that HIPE-1 may work better than HIPE-2, as HIPE-2 works better than HIPE-3 if one can illustrate the benefits of hierarchical explanations. However, 2 reviewers and AE think that this has not been illustrated enough: Table 4 is incomplete missing HIPE-1, and Fig. 6 is not showing the points argued in the rebuttal.

From the AE standpoint, it is a difficult decision to make between major revision and minor revision. However, after deliberation, the AE believes the benefits of hierarchical explanations have indeed not been illustrated enough from the paper from both the qualitative and quantitative standpoint and hence a decision would need to be made after the revision, hence the recommendation of a major revision. The AE believes that once those revisions are provided satisfactorily, this paper can be accepted.

**Audience:**

Yes. Audience that are interested in explainable deep learning.

**Claims And Evidence:**

This paper studies the problem of interpret deep neural networks. Specifically, targeting one type of existing approach that derive prototypes based on dissecting the visual inputs to explain the classification results, hierarchy is introduced upon the prototypes to explain the concepts at different levels. Besides, the authors used hyperbolic space to embed their features into.

**Resubmission Of Major Revision:**

The authors may consider submitting a major revision at a later time.